# Exploring the Role of Voice Assistants in Managing Noncommunicable Diseases: A Systematic Review on Clinical, Behavioral Outcomes, Quality of Life, and User Experiences

**DOI:** 10.3390/healthcare13050517

**Published:** 2025-02-27

**Authors:** Alessia Bramanti, Angelo Corallo, Gennaro Clemente, Luca Greco, Marina Garofano, Massimo Giordano, Claudio Pascarelli, Gianvito Mitrano, Maria Pia Di Palo, Federica Di Spirito, Massimo Amato, Marianna Bartolomeo, Rosaria Del Sorbo, Michele Ciccarelli, Placido Bramanti, Pierluigi Ritrovato

**Affiliations:** 1Department of Medicine, Surgery and Dentistry, University of Salerno, Via S. Allende, 84081 Baronissi, Italy; abramanti@unisa.it (A.B.); mariapia140497@gmail.com (M.P.D.P.); fdispirito@unisa.it (F.D.S.); mamato@unisa.it (M.A.); mbartolomeo@unisa.it (M.B.); rdelsorbo@unisa.it (R.D.S.); mciccarelli@unisa.it (M.C.); 2Department of Engineering for Innovation, University of Salento, 73100 Lecce, Italy; angelo.corallo@unisalento.it (A.C.); claudio.pascarelli@unisalento.it (C.P.); gianvito.mitrano@unisalento.it (G.M.); 3Department of Diabetology, University Hospital “San Giovanni di Dio e Ruggi d’Aragona”, Via San Leonardo, 84125 Salerno, Italy; gennaro.clemente@sangiovannieruggi.it; 4Department of Information Engineering, Electrical Engineering, and Applied Mathematics, University of Salerno, Via Giovanni Paolo II, 132, 84084 Fisciano, Italy; lgreco@unisa.it (L.G.); pritrovato@unisa.it (P.R.); 5Faculty of Psychology, University eCampus, 22060 Novedrate, Italy; bramanti.dino@gmail.com

**Keywords:** voice assistants, non-communicable diseases (NCDs), healthcare technology, artificial intelligence, chronic disease management, digital health tools

## Abstract

Background: Non-communicable diseases (NCDs) represent a leading cause of global mortality, demanding innovative approaches to management. Voice assistants (VAs) have emerged as promising tools in healthcare, offering support for self-management, behavioral engagement, and patient care. This systematic review evaluates the role of VAs in NCD management, analyzing their impact on clinical and behavioral outcomes, quality of life, usability, and user experiences while identifying barriers to their adoption. Methods: A systematic search was conducted in PubMed, Scopus, and Web of Science from January 2014 to October 2024. Studies were selected based on predefined inclusion and exclusion criteria using the PRISMA guidelines. Data extraction focused on outcomes such as usability, acceptability, adherence, clinical metrics, and quality of life. The risk of bias was assessed using the Cochrane Risk of Bias (RoB) 2 and ROBINS-I tools. Results: Eight studies involving 541 participants were included, examining VAs across various NCD contexts such as diabetes, cardiovascular diseases, and mental health. While VAs demonstrated good usability and moderate adherence, their clinical and quality-of-life outcomes were modest. Behavioral improvements, such as increased physical activity and problem-solving skills, were noted in some interventions. Key challenges included privacy concerns, speech recognition errors, and accessibility issues. Conclusions: VAs show potential as supportive tools in NCD management, especially for enhancing patient engagement and self-management, and their impact on clinical outcomes and long-term usability requires further investigation. Future research should focus on diverse populations, standardized metrics, and comparative studies with alternative technologies.

## 1. Introduction

Noncommunicable diseases (NCDs) are responsible for 41 million deaths annually, representing 74% of global mortality [1]. Each year, the deaths related to NCDs account for 17 million before the age of 70, and 86% occur in low- and middle-income countries [1,2]. Cardiovascular diseases are associated with the highest number of deaths among NCDs, amounting to 17.9 million deaths annually, followed by cancers (9.3 million), chronic respiratory diseases (4.1 million), and diabetes (2 million, including kidney disease linked to diabetes) [1,2]. Together, these four diseases account for over 80% of premature deaths caused by NCDs. Risk factors such as tobacco use, physical inactivity, harmful alcohol consumption, unhealthy diets, and air pollution significantly increase the risk of NCD-related deaths [3,4]. Addressing NCDs requires a comprehensive approach that includes early detection, screening, treatment, and palliative care. In light of these considerations, both the healthcare system and patients have begun to benefit from emerging technologies, including voice assistants, particularly in telemedicine and telerehabilitation [5,6] Voice assistant (VAs) gained popularity in commerce due to their usability; in fact, digital voice assistants have become an essential part of everyday life [7]. By 2018, 15.4% of the United States population and 5.9% of the German population owned an Amazon Echo, reflecting the rapid adoption of voice assistants in private households, with smart home purchases increasing by 116% in the third quarter of 2018 compared to the previous year [8].

Beyond commercial applications, VAs have emerged as valuable tools in healthcare providing real-time medication reminders, virtual care, and e-monitoring, enhancing patient engagement and self-management [9]. Studies have demonstrated the reliability of commercial VAs, such as Amazon Alexa, Apple Siri, and Google Assistant, in responding to health-related queries pertinent to NCD management. These VAs can provide accurate information, supporting patients in making informed health decisions, and can be utilized to augment health service delivery, particularly during times when traditional healthcare access may be limited [10,11].

VAs are part of a broader category of conversational agents (CAs), which include artificial intelligence-driven chatbots capable of engaging in dynamic and interactive conversations. Unlike VAs, which primarily rely on voice commands and responses, CAs can incorporate text-based interactions and more advanced dialogue management to provide tailored healthcare support [5,12]. These technologies have gained significant attention in healthcare, supporting telemedicine, self-management of chronic diseases, and mental health interventions [13,14]. Integrating VAs and CAs into healthcare systems has the potential to enhance patient engagement, improve access to health information, and support behavioral change strategies, especially for individuals with NCDs [15].

Elderly individuals frequently face isolation, anxiety, and a feeling of helplessness, both in their homes and in care facilities, which can have a substantial effect on their physical and mental well-being [16]. Speech-based assistants can serve as a valuable tool for individuals who struggle to use other technology-driven services requiring manual dexterity, mobility, or good vision [17]. These systems have the potential to improve the independence of individuals with chronic conditions and enhance their quality of life (QoL), even in the presence of physical or cognitive impairments [18,19]. The aim of this systematic review is to explore the integration of VAs in healthcare, particularly their use in managing NCDs. By analyzing evidence from the included studies, this review aims to assess the impact of VAs on clinical and behavioral outcomes, quality of life, and user experiences, identifying the benefits and challenges associated with their adoption, including usability, acceptability, and readiness to use these technologies. Furthermore, this review provides insights into how VAs contribute to enhancing patient engagement, supporting self-management practices, and addressing broader healthcare needs across different populations.

## 2. Materials and Methods

### 2.1. Study Protocol

This systematic review was performed according to the Preferred Reporting Items for Systematic Review and Meta-Analyses (PRISMA) statement [20]; before starting the literature search and data analysis, the related study protocol was registered in the International Prospective Register of Systematic Reviews (PROSPERO) database of systematic review (identification number: CRD42024604358).

The search strategy, research question, and study selection criteria were designed using the PICO model, with the research question framed as follows [21]:Population (P): Subjects with NCDs;Intervention (I): Voice Assistants for healthcare support;Comparison (C): Digital Twins/Avatars or Textual Chatbots for healthcare support;Outcome (O): Outcomes related to QoL, Cost–benefit, Rehospitalizations, Adherence, Accessibility, and any healthcare outcome measures.

### 2.2. Search Strategy and Study Selection

A literature search was conducted by three reviewers (AB, Massimo Giordano, Marina Garofano) independently, across PubMed/MEDLINE, Scopus, and Web of Science (WOS), systematically searched from January 2014 to October 2024 using the following keywords combined by Boolean operators: voice assistant, virtual assistant, speech assistant, healthcare, health services. The selected keywords were chosen to ensure a comprehensive search strategy, capturing relevant studies regardless of indexing with standardized MeSH terms. This approach maximizes search sensitivity by including various terminologies used to describe voice assistants and digital health technologies, thereby reducing the risk of missing pertinent literature. Complete search strategies are provided in Table 1.

Citations obtained through the literature search were recorded, duplicates were eliminated using EndNote, and titles and abstracts were independently screened by three reviewers (Massimo Giordano, Marina Garofano, AB). Available full texts, compliant with inclusion and exclusion criteria, detailed below, were also independently reviewed for potentially eligible studies (see Figure 1 for the study selection process). Any disagreement between the reviewers was solved by discussion and consensus.

The inclusion criteria were as follows:Source: studies published in the English language from January 2014 to 28 October 2024;Study design: randomized controlled trial (RCT), observational studies, feasibility studies;Study population: subjects with NCDs (no age or gender restrictions);Study intervention: use of a voice assistant;Study outcomes: behavioral and clinical outcomes, quality of life, user experiences (usability, readiness, acceptability), cost-effectiveness, rehospitalizations rate, adherence, accessibility.

The exclusion criteria were as follows:Source: studies published before 2014 and after 28 October 2024;Study intervention: studies that do not involve the use of a voice assistant as the primary intervention;Study outcomes: studies that do not report on at least one of the following outcomes, behavioral and clinical outcomes, quality of life, user experiences (usability, readiness, acceptability), cost-effectiveness, rehospitalization rate, adherence, accessibility, or studies that lack any form of quantitative or qualitative measurement of these outcomes.

### 2.3. Data Extraction

Two authors (Massimo Giordano, AB) independently reviewed the titles and abstracts extracted from the database searches to assess their alignment with the inclusion criteria. In cases where they agreed, studies were either included or excluded based on mutual assessment. When discrepancies arose regarding the inclusion or exclusion of a manuscript based on abstract evaluation, these were resolved through discussion and consensus. If consensus could not be reached, a third reviewer (MPDP) was consulted to make the final decision. The data extraction process was structured based on established methodologies and tailored to the research questions of this review. Extracted information included (a) author, year, country; (b) study design; (c) participants; d sample size, mean age; (e) intervention and control group; (f) outcomes; (g) key results.

This systematic approach ensured a comprehensive and consistent collection of critical data, enabling a thorough synthesis of evidence to address the research questions.

### 2.4. Quality Assessment

The risk of bias in the studies included in this systematic review was assessed by two independent reviewers (Marina Garofano, AB), with assistance from another reviewer (FDS) if necessary in case of disagreement to resolve the issue by discussion and achieve consensus. The Cochrane risk-of-bias tool (RoB 2) [22] was used for the RCTs, evaluating the following domains: bias arising from the randomization process, bias due to deviations from intended interventions, bias due to missing outcome data, bias in the measurement of the outcome, and bias in the selection of the reported result. The risk of bias was classified as “low”, “high”, or ”unclear” (Table 2).

The ROBINS-I Tool [25] was used for the non-RCTs to evaluate the following domains: bias due to confounding, bias in the selection of participants, bias in the classification of interventions, bias due to deviations from intended interventions, bias due to missing data, bias in the measurement of outcomes, and bias in the selection of reported results. The risk of bias for these studies was classified as “low”, “moderate”, or ”high” (Table 3).

## 3. Results

### 3.1. Study Selection and Characteristics

The study selection process followed the PRISMA 2020 guidelines [20]. A total of 1410 records were identified through database searches, including PubMed (66 records), Scopus (427 records), and Web of Science (929 records). After removing 46 duplicate records with EndNote, 1376 records remained for screening. Following the screening, 1348 records were excluded based on relevance, leaving 28 reports for retrieval. All reports were successfully retrieved and assessed for eligibility. Of these, 17 were excluded for various reasons, including the absence of non-communicable diseases [14], being review articles [2], and lacking clinical trials [8]. Ultimately, eight studies met the inclusion criteria and were included in the systematic review. These studies were critically appraised to ensure they aligned with the research objectives and provided relevant data for analysis. This selection process is summarized in the PRISMA flow diagram (Figure 1), and in Table 4, there are the descriptive characteristics of the eight included studies, with a focus on (a) author, year, country; (b) study design; (c) participants; (d) sample size, mean age; (e) intervention and control group; (f) outcomes; (g) key results.

**Figure 1 healthcare-13-00517-f001:**
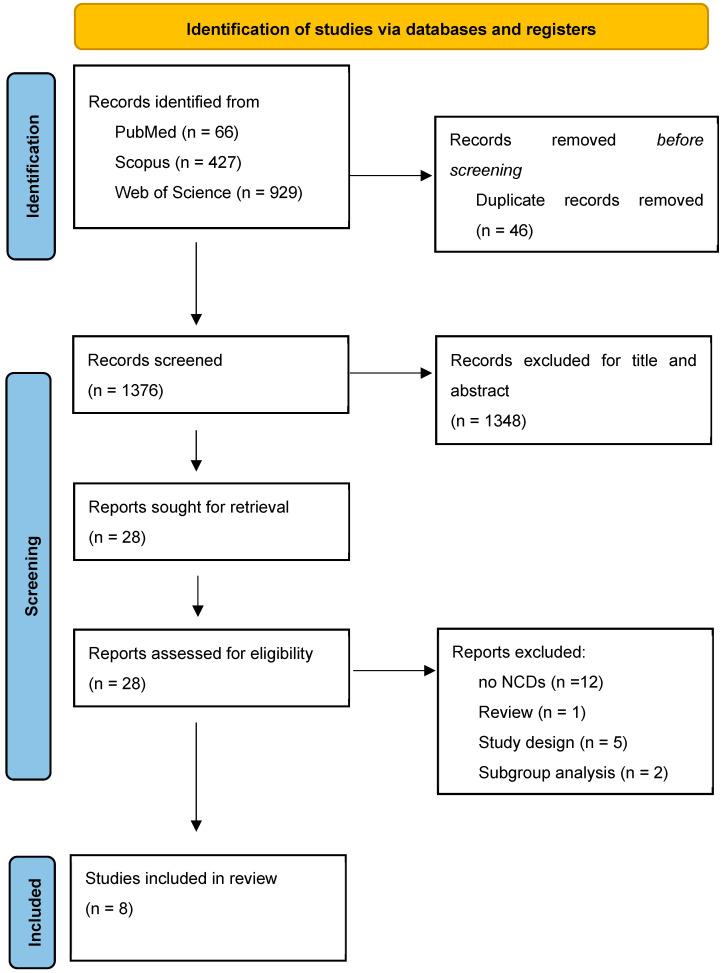
Flow diagram of study selection.

### 3.2. Participant Demographics

The total sample consisted of 541 participants, including diverse populations:

Diabetes management: 178 participants across four studies [23,26,27,30];

Cardiovascular diseases and heart failure (HF): 257 participants across two studies [28,29];

Depression and anxiety: 63 participants across one study [24,28];

Intellectual disabilities: 44 participants across one study [31].

Participants varied in age, gender, and baseline health status, but all the studies included NCDs management with VAs.

### 3.3. Outcome Measures

Studies specifically investigating cost–benefit analyses, rehospitalizations, and accessibility concerning the use of VAs in the management of individuals with NCDs were not identified in the literature. The studies included in this review assessed a variety of outcomes, grouped into behavioral measures, clinical and medical outcomes, quality of life, usability, acceptability, readiness, and adherence (Table 4).

#### 3.3.1. Behavioral Measures

Two studies analyzed behavioral changes, including physical activity and problem-solving skills. For example, in an RCT by Glavas et al. [23], participants in the intervention group showed a significant reduction in sedentary time (−67 min/day, *p* = 0.006) and an increase in moderate activity (+24.7 min/day, *p* = 0.04) compared to the control group. Kannampallil et al. [24] reported minor improvements in problem-solving behaviors, measured through problem-solving indices, with small effect sizes and limited clinical significance (Table 5).

#### 3.3.2. Clinical and Medical Outcomes

Four studies evaluated clinical outcomes such as Hemoglobin A1c (HbA1c) levels, depressive symptoms, and neural activity changes. Baptista et al. [27] observed a slight reduction, not statistically significant, in HbA1c levels from 7.3% ± 1.5 at baseline to 7.1% ± 1.4 at 6-month follow-up (*n* = 66), also Glavas et al. [23] observed potential benefits of the VA on glycemic management with a moderate effect size. Roca et al. [30] reported significant improvements in both HbA1c (*p* = 0.02) and depressive symptoms (*p* = 0.002), with high medication adherence (MPR ≥ 100% for several participants). Kannampallil [24] et al. examined neural activity changes but found only minor, statistically insignificant differences (Table 6).

#### 3.3.3. Quality of Life

Three studies assessed quality of life using standardized tools such as the EQ-5D-5L, WEBWMS, and custom surveys. Glavas et al. (2024) found no significant differences in overall QoL scores but noted slight improvements in the intervention group’s visual analog scale (VAS) ratings (79.2 ± 19.1 to 79.6 ± 21.7). Smith et al. (2023) reported that 80% of participants with intellectual disabilities felt more independent after using voice assistants, even though no significant improvements were observed in well-being scores as measured by WEBWMS. Kannampallil et al. [24] evaluated changes in positive and negative affect as proxies for quality of life. The Positive Affect Score showed a slight increase in the intervention group (from 25.21 ± 6.26 to +4.83 ± 7.79), compared to the control group (+2.43 ± 7.89), but with a negligible effect size (Cohen’s d = 0.1). Negative Affect Scores decreased identically in both groups (−9.07 ± 7.58 in the intervention group and −9.07 ± 5.56 in the control group, Cohen’s d = 0.1). No significant changes were observed in worry levels as measured by the Penn State Worry Questionnaire (PSWQ), with a reduction in the intervention group (−3.95 ± 11.01) and no change in the control group (0.0 ± 10.95), yielding a Cohen’s d of 0.0 (Table 7)

#### 3.3.4. Usability

Six studies evaluated the usability of digital and voice-assisted technologies, providing insights into user experiences and challenges. Usability was generally rated positively, with System Usability Scale (SUS) scores ranging from 70.4 to 92, indicating good to excellent usability.

The “Vitória” virtual assistant for diabetes management [26] received SUS scores of 76.59 from end users and 70.2 from experts, highlighting its simplicity and ease of use; similarly, the “Laura” app [27] was found helpful by 86% of users, with a moderate user engagement (participants interacted with the app 18–36 times over the study period). In contrast, the “Medly” voice app for heart failure management [28] achieved an SUS score of 92/100 with 75% of users that prefer it over traditional methods, but 25% expressed privacy concerns, highlighting the need for better data management.

Amazon Alexa paired with the “Buddy Link” software [23] scored 70.4 on the SUS, reflecting good usability overall; nonetheless, some users faced challenges with specific interface elements, underscoring the variability in user experiences. Among individuals with intellectual disabilities, voice assistants like Amazon Echo and Google Home [31] were rated as easy to use by 73% of participants; however, 41% required frequent assistance, and 25% experienced frustration due to speech recognition issues. Despite these difficulties, 79% of users enjoyed using the devices and continued to engage with them. Among these, a virtual assistant for medication and reminder management [30] demonstrated consistent engagement, with 74.4% of reminders answered and 69% of users planning continued use, despite occasional comprehension issues (2.6%). Retention was high (77%), and older adults particularly appreciated its ease of use, reinforcing the importance of accessibility in digital health solutions (Table 8).

#### 3.3.5. Acceptability and Readiness

Four studies assessed the acceptability and readiness to adopt digital and voice-assisted technologies, highlighting overall positive perceptions and areas for improvement.

Baptista et al. [27] explored user satisfaction with the “Laura” app for diabetes management. The study found that 86% of participants considered the app helpful and friendly, and 73% expressed trust in the virtual assistant. However, some users experienced frustration due to mismatched verbal and nonverbal cues, which limited the overall user experience.

Kowalska et al. [29] investigated readiness for telemedicine and voice technology in cardiovascular patients. The study reported high readiness rates, with 83.9% of participants open to telemedicine and 66.7% willing to use voice technology. This readiness was particularly pronounced among individuals who had faced barriers to healthcare access and was influenced by factors such as higher education levels, urban residence, and strong family support.

Smith et al. [31] focused on individuals with intellectual disabilities using voice assistants like Amazon Echo and Google Home. The study revealed that 79% of participants enjoyed using the devices despite occasional frustration with speech intelligibility. However, 41% required frequent assistance, underscoring the importance of adequate training and support to maximize usability and satisfaction.

Barbaric et al. [28] evaluated the acceptability of the “Medly” app for heart failure management, with 75% of users preferring it over traditional methods (Table 9).

#### 3.3.6. Adherence

Adherence, defined as the level of engagement with voice assistants, was reported in only one study. Smith et al. [31] provided data showing that 57 out of 63 participants actively used the devices, while 6 did not engage with any features. Music was the most frequently used feature (~90%), followed by reminders and weather updates (~40%). Perseverance in using the devices was high, with 79% of participants continuing to use the voice assistants despite challenges such as intelligibility issues or phrasing commands.

Overall, the reviewed interventions demonstrated promising outcomes in usability, behavioral engagement, adherence, and clinical metrics, but challenges such as user training, privacy concerns, and occasional frustration with interfaces were identified, emphasizing the need for iterative design improvements and tailored implementation strategies (Table 10).

## 4. Discussion

This systematic review highlights the potential and challenges of using voice VAs in managing NCDs, emphasizing their role in supporting behavioral engagement, clinical outcomes, and usability, while pointing out the need for improvements in accessibility, privacy, and personalization [32]. An initial aim of this review was to compare the effectiveness of VAs with other types of conversational agents (CAs) in NCD management. However, the current literature lacks studies that directly perform such comparisons and investigating this aspect in future research could provide valuable insights into user preferences and inform strategies to enhance adherence to these technologies. As noted in recent reviews [33,34], for example, incorporating anthropomorphic and context-aware features in conversational agents may strengthen relational outcomes and foster greater user adherence. Investigating these aspects further could guide the development of more tailored and effective interventions.

Furthermore, an important aspect of VA implementation in healthcare is their specific functionalities and regulatory approval status. The applications included in this review demonstrate a range of approaches to NCD management, from symptom tracking and behavior change [23,24,26] coaching to medication reminders [26,30] and clinician alerts [26,28]. For example, Amazon Alexa and Echo [23,29,35] integrates with wearable devices and mobile health apps to provide personalized lifestyle tracking and health coaching, making it a flexible tool for managing diabetes and obesity and cardiovascular disease. Similarly, Medly Voice Assistant [28] is specifically tailored for heart failure management, enabling remote symptom tracking, daily patient feedback, and clinician alerts when deterioration is detected. Other applications, such as Vitória and Laura [26,27], focus on diabetes self-management, offering medication reminders, dietary coaching, and emotional support through conversational artificial intelligence. In addition, the Signal-based virtual assistant studied by Roca et al. [30] is designed to enhance medication adherence in patients with type 2 diabetes and depressive disorder by providing structured reminders and enabling clinician monitoring through a secure messaging platform (Table 11).

A key aspect emerging from this systematic review is the lack of explicit mention of regulatory approvals, such as FDA or MDR CE certification, in the included studies. None of the analyzed voice assistants were reported to have undergone regulatory approval processes, raising concerns about their compliance with established medical device regulations. This omission suggests that many of these technologies may not yet meet the safety, efficacy, and data protection standards required for clinical use.

The absence of regulatory approval may be attributed to several factors. First, some of the voice assistants examined in this review are research prototypes or commercially available AI-driven tools that have been repurposed for healthcare applications rather than specifically designed as certified medical devices. Second, the regulatory classification of voice assistants in healthcare remains an evolving area, and many interventions may not yet fall under the category of Software as a Medical Device (SaMD), thus operating in a regulatory gray zone.

This is further supported by the limited number of studies available, with only eight included in this review, all involving a small patient population. Given the growing emphasis on regulatory compliance for Software as a Medical Device (SaMD), future research should investigate how voice assistants can meet FDA and MDR requirements and explore strategies to ensure their clinical safety and effectiveness.

Regarding clinical and behavioral outcomes, the reviewed studies showed modest improvements. For instance, Roca et al. [30] observed better glycemic control and reduced depressive symptoms using a VA intervention. These findings align with broader evidence, suggesting that personalized conversational agents can deliver behavior change strategies effectively, as noted by Anisha et al. [33], who highlighted their role in promoting self-management and behavioral modifications for NCDs. However, some interventions, such as Kannampallil et al. [24], reported limited clinical impact, emphasizing the importance of targeting intervention design to user needs. A promising opportunity for delivering tailored interventions lies in the integration of artificial intelligence that enhances personalization with more precise recommendations and interventions with a positive impact on clinical and behavioral outcomes [36].

User experience and usability are investigated in most of the studies included in this review, highlighting their critical importance when discussing virtual assistants. This focus underscores the central role that ease of use, interface design, and user satisfaction play in determining the effectiveness and adoption of these technologies. In the included studies, usability scores ranged from moderate to excellent [23,26,27,28], and participants frequently reported ease of use but highlighted challenges such as speech recognition errors [31] and interface complexities [26]. For elderly users, usability issues can pose significant barriers, particularly when dealing with complex systems that fail to account for age-related sensory or cognitive impairments [37]. Anisha et al. [33] also identified usability as a key determinant of success for conversational agents, particularly in populations with low health literacy, and Sawad et al. [38] highlighted that user satisfaction with CAs often stems from their ability to provide nonjudgmental, easily accessible support. However, some users found certain embodied agents annoying or difficult to engage with. Improved designs incorporating adaptive learning and anthropomorphic features may further enhance usability and user trust.

Finally, adherence to VA interventions was generally moderate. For instance, Smith et al. [31] reported active engagement from 90% of users, with perseverance levels high, despite challenges that underline the need for improved user-centric design, tailored support systems, and rigorous testing to enhance the effectiveness and adherence to such interventions in diverse populations.

### 4.1. Conclusions

This systematic review underscores the potential of VAs as an innovative tool in managing NCDs, offering diverse benefits across clinical, behavioral, and usability domains. While VAs demonstrate promise in promoting self-management, enhancing patient engagement, and improving usability scores, their impact on clinical and quality-of-life outcomes remains modest, reflecting variability in user experiences and intervention designs, also with privacy concerns, speech recognition errors, and accessibility challenges that limit widespread adoption. Future research should focus on including larger, diverse populations to improve the generalizability of findings and ensure underrepresented groups are adequately studied, such as those with low health literacy or limited technological access. Employing more rigorous study designs, such as multicenter RCTs, can provide stronger evidence for the effectiveness of VAs. The adoption of standardized and validated outcome measures across studies will enable better comparisons and synthesis of results.

### 4.2. Limitations and Research Gaps

This review underscores several limitations that need to be addressed to optimize the effectiveness and adoption of voice assistants in healthcare. First of all, many of the included studies involved small participant groups, which limits the generalizability of findings. Observational and pilot studies formed a significant portion of the reviewed literature, reducing the ability to draw robust causal conclusions; secondly, the studies measured diverse outcomes ranging from usability and adherence to clinical and behavioral improvements, making direct comparisons challenging, and also the lack of standardized metrics further complicates synthesizing results. Finally, the majority of studies focused on specific demographic groups, such as adults with diabetes or cardiovascular diseases. Vulnerable populations, including those with lower health literacy, limited access to technology, or residing in rural areas, were underrepresented.

## Figures and Tables

**Table 1 healthcare-13-00517-t001:** Search Strategy.

Database	Search Terms	Filters Applied	Date of Search
PubMed	“Voice Assistant” OR “Virtual Assistant” OR “Vocal Assistant” OR “Speech Assistant” OR “Voice-Activated Assistant” OR “AI Assistant” OR “Digital Assistant” OR “Conversational Agent” OR “Intelligent Personal Assistant” OR “Smart Assistant” OR “Speech Recognition System” AND “Healthcare” OR “Health Services” OR “Health Care Quality” OR “Public Health” OR “Health Care” OR “Health Policy”	Publication years: 2014–2024, Article type: RCT, Clinical Trial, Species: Humans, Language: English, Age: 19+	28 October 2024
Scopus	“Voice Assistant” OR “Virtual Assistant” OR “Vocal Assistant” OR “Speech Assistant” OR “AI Assistant” OR “Digital Assistant” AND “Healthcare” OR “Health Services” OR “Public Health”	Publication years: 2014–2024, Article type: Research articles, Others	28 October 2024
Web of Science	“Voice Assistant” OR “Virtual Assistant” OR “Vocal Assistant” OR “Speech Assistant” OR “Voice-Activated Assistant” OR “AI Assistant” OR “Digital Assistant” OR “Conversational Agent” OR “Intelligent Personal Assistant” OR “Smart Assistant” OR “Speech Recognition System” AND “Healthcare” OR “Health Services” OR “Health Care Quality” OR “Public Health” OR “Health Care” OR “Health Policy”	Publication years: 2014–2024, Document types: Article, Language: English	28 October 2024

**Table 2 healthcare-13-00517-t002:** Cochrane risk of bias tool for the risk of bias in individual studies.

	Glavas C. et al., 2024 [23]	Kannampallil T. et al., 2024 [24]
Bias arising from the randomization process	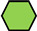	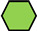
Bias due to deviations from intended interventions	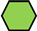	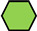
Bias due to missing outcome data	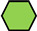	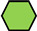
Bias in measurement of the outcome	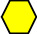	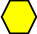
Bias in selection of the reported result	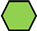	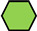
Overall	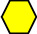	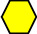

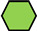
: Low Risk Of Bias; 
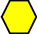
: Unclear Risk Of Bias; 
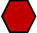
: High Risk Of Bias.

**Table 3 healthcare-13-00517-t003:** ROBINS-I Tool for non-RCTs; abbreviations: PY (Probably Yes), P (Possibly), NY (Probably No), N (No).

Article	Bias Due to Confounding	Bias in Selection of Participants	Bias in Classification of Interventions	Bias Due to Deviations from Intended Interventions	Bias Due to Missing Data	Bias in Measurement of Outcomes	Bias in Selection of Reported Results	Overall Risk of Bias
Balsa J. et al., 2019 [26]	PY	P	P	P	NY	PY	N	MODERATE
Baptista S. et al., 2020 [27]	PY	P	P	NY	NY	PY	N	MODERATE
Barbaric A. et al., 2022 [28]	PY	P	P	NY	N	PY	N	MODERATE
Kowalska M. et al., 2020 [29]	PY	P	P	NY	NY	PY	N	MODERETE
Roca S. et al., 2021 [30]	PY	P	P	NY	NY	PY	N	MODERATE
Smith E. et al., 2023 [31]	PY	P	P	P	NY	PY	N	MODERATE

**Table 4 healthcare-13-00517-t004:** Descriptive characteristics of the included studies.

Author, Year, Country	Study Design	Participants	Sample Size, Mean Age	Intervention and Control Group	Outcome Measures	Key Results
Balsa J. et al., 2020, Portugal [26]	Observational	Older adults with Type 2 diabetes	N = 20 (11 users: 3 F, 8 M; 9 experts: 8 F, 1 M)	Virtual assistant ‘Vitória’ for medication adherence, physical activity, and diet. Uses behavior change techniques.	- Usability: SUS, qualitative feedback - Acceptability: Feedback on user experience	- Usability: SUS: 76.6 (users), 70.2 (experts) (Good–excellent usability) - Acceptability: Positive aspects: easy to use; Suggested improvements: reduce repetitions, better interface, more features
Baptista S. et al., 2020, Australia [27]	Mixed methods	Adults with Type 2 diabetes	N = 93 (49 M, 44 F), mean age 55	ECA-based app (“Laura”) for diabetes self-management, emotional support, and education.	- Clinical or Medical Outcomes: HbA1c - Usability: Survey - Acceptability: Survey	- Clinical or Medical Outcomes: decrease in HbA1c levels ↓ (7.3% → 7.1%) - Acceptability: 86% found it useful, 44% felt motivated, 20% frustrated - Usability: Issues: monotonous voice, mismatched gestures
Barbaric A. et al., 2022, Canada [28]	Observational	Patients with HF	N = 8	A voice app version of Medly for HF management, daily monitoring, and feedback.	- Usability: SUS, interviews - Acceptability: Preferences for voice app vs. smartphone	- Usability: SUS: 92/100 (Excellent usability) - Acceptability: 75% preferred voice app over smartphone, 25% had privacy concerns
Glavas C. et al., 2024, Australia [23]	RCT	Adults with obesity and diabetes	N = 50 (29 M, 21 F); IG = 25 (mean age 65), CG = 25 (mean age 67.3)	IG: Alexa Echo Show 8 + “Buddy Link” for personalized exercise and diet reminders. CG: Generic physical activity and diet info via email.	- Behavioral Measures: Physical activity (accelerometer) - Clinical or Medical Outcomes: Diabetes self-care (DSMQ) - Quality of Life: EQ-5D-5L - Usability: SUS	- Behavioral Measures: Decrease in Sedentary time ↓67 min/day (*p* = 0.006), Increase in Moderate activity ↑24.7 min/day (*p* = 0.04) - Usability: SUS: 70.4/100 (Good usability) - Quality of Life: No significant changes
Kannampallil T. et al., 2023, USA [24]	Pilot RCT	Adults with mild-to-moderate depression/anxiety	N = 63 (20 M, 43 F); IG = 42, CG = 21, mean age 37.8	IG: ‘Lumen’ voice coach on Alexa for Problem-Solving Therapy (PST) with 8 sessions and reminders. CG: Waitlist control.	- Behavioral Measures: Problem-solving (SPSI-R:S, PPO, NPO, RPS, ICS, AS) - Clinical or Medical Outcomes: Neural markers (Amygdala, dlPFC) - Quality of Life: Penn State Worry, affect scores - Acceptability: Dysfunctional Attitudes Scale	- Behavioral Measures: Small improvements in problem solving - Clinical or Medical Outcomes: Limited neural changes - Acceptability: ↓ Decrease in Dysfunctional attitudes (d = 0.6, moderate)
Kowalska M. et al., 2020, Poland [29]	Observational (cross-sectional)	Patients with cardiovascular diseases	N = 249 (158 M, 91 F), mean age 65.3	Voice assistants + telemedicine services for remote cardiologist contact and monitoring.	- Readiness: Survey on telemedicine and voice assistant adoption	- Readiness: 83.9% readiness for telemedicine, 66.7% for voice technology - Key factors: prior healthcare access issues, urban living, higher education
Roca S. et al., 2021, Spain [30]	Observational	Patients with diabetes mellitus and depressive disorders	N = 13 (9 F, 4 sM), mean age 63.8	Virtual assistant on the Signal platform for medication reminders, appointment scheduling, and feedback on adherence.	- Clinical or Medical Outcomes: HbA1c, PHQ-9 - Adherence: Medication adherence - Usability: Acceptance, real use	- Clinical or Medical Outcomes: HbA1c ↓ (*p* = 0.02), PHQ-9 ↓ (*p* = 0.002) - Adherence: 74.4% responded to reminders - Acceptability: 69% planned continued use
Smith E. et al., 2023, UK [31]	Mixed-methods semi-RCT	Individuals with mild-to-moderate intellectual disabilities	N = 44 (IG = 22, CG = 22); IG mean age 45.3, CG mean age 48.6	IG: Alexa/Google Home for increased independence and well-being. CG: No device provided.	- Quality of Life: WEBWMS - Behavioral Measures: Independence - Usability: Ease of use, challenges - Readiness: Technology awareness, training needs - Acceptability: User satisfaction, frustration - Adherence: Feature utilization, perseverance	- Quality of Life: No significant improvement - Behavioral Measures: 80% felt more independent - Usability: 73% found it easy, but 41% needed frequent assistance - Acceptability: 79% enjoyed it, 25% frustrated (speech issues) - Adherence: Most-used feature: music (90%)

Abbreviations: AS, Avoidance Style; BCTs, Behavior Change Techniques; CG, Control Group; DAS, Dysfunctional Attitudes Scale; dlPFC, Dorsolateral Prefrontal Cortex; DSMQ, Diabetes Self-Management Questionnaire; ECA, Embodied Conversational Agent; EQ-5D-5L, EuroQol-5 Dimensions-5 Levels; F, female; HbA1c, Hemoglobin A1c; IG, Intervention Group; ICS, Impulsivity in Problem Solving; M, male; N, number; NPO, Negative Problem Orientation; PHQ-9, Patient Health Questionnaire-9; PPO, Positive Problem Orientation; PST, Problem-Solving Therapy; RPS, Rational Problem Solving; SPSI-R:S, Social Problem-Solving Index-Revised Short Form; SUS, System Usability Scale; WEBWMS, Warwick–Edinburgh Mental Wellbeing Scale.

**Table 5 healthcare-13-00517-t005:** Behavioral measures.

Study	Behavioral Outcome	Measurement Tool	Key Findings
Glavas et al., 2024 [23]	Physical activity	ActiGraph GT9XLink (accelerometer)	↓ Decrease in Sedentary time: -67 min/day (*p* = 0.006) ↑ Increase in Mod. activity: +24.7 min/day (*p* = 0.04) ↑ Increase in MVPA: +30.9 min/day (*p* = 0.046)
Kannampallil et al., 2023 [24]	Problem-solving skills	SPSI-R:S, PPO, NPO, RPS, ICS, AS	Minor improvements, Cohen’s d = 0.0–0.3 No clinically meaningful differences

Abbreviations: MVPA, Moderate to Vigorous Physical Activity; SPSI-R:S, Social Problem-Solving Index-Revised Short Form; PPO, Positive Problem Orientation; NPO, Negative Problem Orientation; RPS, Rational Problem Solving; ICS, Impulsivity in Problem Solving; AS, Avoidance Style.

**Table 6 healthcare-13-00517-t006:** Clinical and medical outcomes.

Study	Clinical Outcome	Measurement Tool	Key Findings
Baptista et al., 2020 [27]	HbA1c	Lab tests	↓ Decrease in HbA1c levels: 7.3% ± 1.5 → 7.1% ± 1.4 at 6M (n = 66) Interviewed patients: 6.8% ± 0.9
Glavas et al., 2024 [23]	Diabetes self-care	DSMQ	Moderate effect size, not significant
Roca et al., 2021 [30]	HbA1c, depressive symptoms, medication adherence	HbA1c, PHQ-9, MPR	↓ HbA1c (*p* = 0.02) ↓ PHQ-9 (*p* = 0.002) MPR ≥ 100% in several pts
Kannampallil et al., 2023 [24]	Neural activation	fMRI	No significant changes

Abbreviations: HbA1c, Glycosylated Hemoglobin; DSMQ, Diabetes Self-Management Questionnaire; PHQ-9, Patient Health Questionnaire-9; MPR, Medication Possession Ratio; fMRI, Functional Magnetic Resonance Imaging.

**Table 7 healthcare-13-00517-t007:** Quality of life (QoL) outcomes.

Study	QoL Measure	Measurement Tool	Key Findings
Glavas et al., 2024 [23]	General QoL	EQ-5D-5L, VAS	No significant changes Slight increase in ↑ VAS score: IG: 79.2 → 79.6, CG: 70.6 → 72.9
Smith et al., 2023 [31]	Well-being, independence	WEBWMS, custom survey	80% felt more independent No significant change in WEBWMS
Kannampallil et al., 2023 [24]	Emotional well-being	PA, NA Scores	↑ Increase in PA: +4.83 (IG) vs. +2.43 (CG), Cohen’s d = 0.1 ↓ Decrease in NA: −9.07 both groups (Cohen’s d = 0.1)

Abbreviations: QoL, Quality of Life; EQ-5D-5L, EuroQol-5 Dimensions-5 Levels; VAS, Visual Analog Scale; WEBWMS, Warwick–Edinburgh Mental Well-Being Scale; PA, Positive Affect; NA, Negative Affect; IG, Intervention Group; CG, Control Group.

**Table 8 healthcare-13-00517-t008:** Usability outcomes.

Study	Usability Measure	Measurement Tool	Key Findings
Balsa et al., 2019 [26]	Usability	SUS	SUS: 76.59/100 (end-users), 70.2/100 (experts) Feedback: UI issues (small buttons, dialogue repetitions)
Baptista et al., 2020 [27]	User feedback	Survey	86% found helpful, but with issues: monotone voice, gesture mismatch
Barbaric et al., 2022 [28]	Usability	SUS	SUS: 92/100 75% preferred VA over smartphone
Glavas et al., 2024 [23]	Usability	SUS	SUS: 70.4/100, high variability (SD = 16.9)
Roca et al., 2021 [30]	Usability	Acceptance and real use of the virtual assistant.	Daily interactions: 2.7/day (88.5% numeric-based); 74.4% of reminders answered; 77% retention (23% uninstalled); 69% planned continued use. Older adults noted ease of use despite occasional challenges.
Smith et al., 2023 [31]	Usability	Ease of Use: Likert-scale survey and staff observations. Challenges: Open-ended feedback and frustration ratings.	A total of 73% easy to use, 79% enjoy to use, 41% needed assistance, 25% had frustration (speech recognition issues).

Abbreviations: SUS, System Usability Scale; UI, User Interface; VA, Virtual Assistant.

**Table 9 healthcare-13-00517-t009:** Acceptability and readiness outcomes.

Study	Acceptability/Readiness Measure	Measurement Tool	Key Findings
Baptista et al., 2020 [27]	Acceptability	Survey	86% helpful, 85% competent, 73% trust VA
Kowalska et al., 2020 [29]	Readiness	Survey	83.9% open to telemedicine, 66.7% willing to use VA
Smith et al., 2023 [31]	Acceptability and Readiness	Pre-intervention survey (Likert-scale) for readiness; user satisfaction survey (Likert-scale) on enjoyment for acceptability	79% enjoyed VA use, 41% needed assistance
Barbaric et al., 2022 [28]	Acceptability	Survey	75% preferred Medly VA over phone

Abbreviations: VA, Virtual Assistant.

**Table 10 healthcare-13-00517-t010:** Adherence outcomes.

Study	Adherence Measure	Measurement Tool	Key Findings
Smith et al., 2023 [31]	Engagement with VA, feature utilization, perseverance	Self-reported usage and engagement data	A total of 57/63 participants actively used the devices; 6/63 did not engage with any features; Music was the most used feature (~90%); Reminders and weather updates were used by ~40%; 79% continued using VA despite challenges.

Abbreviations: VA, Virtual Assistant.

**Table 11 healthcare-13-00517-t011:** Comparative overview of voice assistant applications.

Application	Disease	Key Features	AI Capabilities	User Interaction	FDA/MDR CE Approval
Amazon Alexa, Amazon Echo	Diabetes, Obesity, CVD	Personalized coaching, medication reminders, lifestyle tracking	Natural Language Processing (NLP), integration with wearables	Voice-based	Not specified
Medly	CVD	Symptom tracking, clinician alerts, remote monitoring	AI-driven alerts, symptom analysis	Voice + app	Not specified
Vitória	Type 2 Diabetes	Medication adherence, dietary support, behavior change	Behavior Change Techniques (BCTs), patient feedback	Voice-based	Not specified
Laura	Type 2 Diabetes	Emotional support, diabetes education, self-management tools	Avatar-based interactions, NLP	Voice + text	Not specified
Lumen	Mental Health	Cognitive Behavioral Therapy (CBT)-based problem solving	AI-driven conversation, NLP-based coaching	Voice-based	Not specified
Signal Platform	Type 2 Diabetes, Depressive Disorder	Medication reminders, clinician monitoring, patient self-reporting	AI-assisted chatbot or call-based structured messaging	Text- or call-based	Not specified

Abbreviations: AI, Artificial Intelligence; BCTs, Behavior Change Techniques; CA, Conversational Agent; CBT, Cognitive Behavioral Therapy; CE, Conformité Européenne; CVD, Cardiovascular Disease; FDA, Food and Drug Administration; MDR, Medical Device Regulation; NCD, Non-Communicable Disease; NLP, Natural Language Processing; SaMD, Software as a Medical Device; VA, Virtual Assistant.

## Data Availability

All data are included in this study.

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
