# Peer review of "Exploring the Role of Voice Assistants in Managing Noncommunicable Diseases: A Systematic Review on Clinical, Behavioral Outcomes, Quality of Life, and User Experiences"

_healthcare, 2025, doi:10.3390/healthcare13050517_

Round 1

Reviewer 1 Report

Comments and Suggestions for Authors

This paper presents a systematic review on integrating virtual assistants (VAs) in the management of non-communicable diseases (NCDs). The focus is on evaluating the resul of using VAs on clinical and behavioral outcomes, quality of life, and overall user experience. 

The review follows the PRISMA protocol and gathers publications from January 2015 to October 2024, with several combinations between relevant search terms, in PubMed/MEDLINE, Scopus, and Web of Science (WOS). 

The inclusion and exclusion criteria are stated and explained: studies from January 2015 to October 2024, that use Randomized Control Trial (RCT), observational or feasability studies, subjects with NCDs without age/gender restrictions. The use of a VA is mandatory, and the outcomes of the study must include clinical and behavioral outcomes, quality of life, and overall user experience. 

The risk of bias for RCTs was evaluated by two independent reviewers, whereas a third reviewer was involved in case of disagreement, using the Cochrane Risk-of-Bias Tool. The ROBINS-I Tool was used for non-RCTs. 

The identification, screening and inclusion procedure is explained via the flow diagram of the PRISMA protocol. Finally, 8 studies were included in the review. 

The results of analysing the 8 studies are explained in separate sections: behavioral measures, clinical and medical outcomes, quality of life, usability, acceptability, readiness and adherence, and summarized in a table. For each of these outcomes, the authors present the results in the studies that included such measurements. It is found that most studies evaluate user experience and usability, while the level of engagement with VAs, or adherence, was addressed in a single study. 

On important result of the review is discovering limitations and research gaps: size of participant groups, variety of measured outcomes that limits direct comparisons, lack of standardized metrics, lack of variety in demographic groups. 

The paper is well structured and clear. Some improvements can be made when presenting the outcome measures, to enhance readability and understanding -- I would suggest some tables and/or graphic representations of the values in Section 3.3. 
Table 4 is very difficult to follow, consider reformatting and including smaller sections of it in the main manuscript. 

Author Response

REV1

This paper presents a systematic review on integrating virtual assistants (VAs) in the management of non-communicable diseases (NCDs). The focus is on evaluating the resul of using VAs on clinical and behavioral outcomes, quality of life, and overall user experience. 

The review follows the PRISMA protocol and gathers publications from January 2015 to October 2024, with several combinations between relevant search terms, in PubMed/MEDLINE, Scopus, and Web of Science (WOS). 

The inclusion and exclusion criteria are stated and explained: studies from January 2015 to October 2024, that use Randomized Control Trial (RCT), observational or feasability studies, subjects with NCDs without age/gender restrictions. The use of a VA is mandatory, and the outcomes of the study must include clinical and behavioral outcomes, quality of life, and overall user experience. 

The risk of bias for RCTs was evaluated by two independent reviewers, whereas a third reviewer was involved in case of disagreement, using the Cochrane Risk-of-Bias Tool. The ROBINS-I Tool was used for non-RCTs. 

The identification, screening and inclusion procedure is explained via the flow diagram of the PRISMA protocol. Finally, 8 studies were included in the review. 

The results of analysing the 8 studies are explained in separate sections: behavioral measures, clinical and medical outcomes, quality of life, usability, acceptability, readiness and adherence, and summarized in a table. For each of these outcomes, the authors present the results in the studies that included such measurements. It is found that most studies evaluate user experience and usability, while the level of engagement with VAs, or adherence, was addressed in a single study. 

On important result of the review is discovering limitations and research gaps: size of participant groups, variety of measured outcomes that limits direct comparisons, lack of standardized metrics, lack of variety in demographic groups. 

The paper is well structured and clear. Some improvements can be made when presenting the outcome measures, to enhance readability and understanding -- I would suggest some tables and/or graphic representations of the values in Section 3.3. 
Table 4 is very difficult to follow, consider reformatting and including smaller sections of it in the main manuscript. 

Thank you for your comment, we appreciate the reviewer's insightful comments and constructive feedback on our systematic review on virtual assistants in the management of non-communicable diseases. Below, we address each point raised and outline the revisions made to improve the manuscript.

Clarification and Enhancement of Outcome Measures Presentation: we acknowledge the reviewer's suggestion to improve the readability and understanding of the outcome measures. To address this, we have revised Section 3.3 by incorporating additional, these visual elements enhance clarity by summarizing key findings concisely and improving comparability across studies.

Reformatting Table 4: we agree that Table 4 was difficult to follow in its previous format. We have restructured it into smaller, more digestible format, ensuring better readability and alignment with the text.

Reviewer 2 Report

Comments and Suggestions for Authors

This systematic review assesses the significance of voice assistants (VAs) in the management of non-communicable diseases (NCDs) by examining their effects on clinical and behavioral outcomes, quality of life, usability, and user experiences, as well as identifying obstacles to their adoption. The review covers publications between January 2015 and October 2024 in Web of Science, PubMed, and Scopus. The systematic review included eight studies where users assessed VAs in a variety of NCD situations. VAs had limited clinical and quality-of-life outcomes, despite their strong usability and moderate adherence. Some therapies showed behavioral changes, including greater physical activity and improved problem-solving abilities. Accessibility limitations, voice recognition mistakes, and privacy concerns were major obstacles. The review suggest that further research is needed to determine how VAs affect clinical results and long-term usability. 

The manuscript will be of great interest to the readers of this journal. It is well-written. However, the authors should consider the following comments/suggestions to improve the overall quality of the manuscript. 

1. Lines 181 - 184: The numbers listed in these statements do not match those shown in Figure 1. These statements indicate that 11 studies were included in this systematic study. However, Figure 1 as well as the other sections of the manuscript indicate 8 studies to be included in the study. Check and update accordingly.

2. There are spelling errors in the manuscript.  The authors should use a spell-checker for the entire manuscript. 

3. Define all acronyms at first instance of usage. 

Author Response

REV2

This systematic review assesses the significance of voice assistants (VAs) in the management of non-communicable diseases (NCDs) by examining their effects on clinical and behavioral outcomes, quality of life, usability, and user experiences, as well as identifying obstacles to their adoption. The review covers publications between January 2015 and October 2024 in Web of Science, PubMed, and Scopus. The systematic review included eight studies where users assessed VAs in a variety of NCD situations. VAs had limited clinical and quality-of-life outcomes, despite their strong usability and moderate adherence. Some therapies showed behavioral changes, including greater physical activity and improved problem-solving abilities. Accessibility limitations, voice recognition mistakes, and privacy concerns were major obstacles. The review suggest that further research is needed to determine how VAs affect clinical results and long-term usability. 

The manuscript will be of great interest to the readers of this journal. It is well-written. However, the authors should consider the following comments/suggestions to improve the overall quality of the manuscript. 

  1. Lines 181 - 184: The numbers listed in these statements do not match those shown in Figure 1. These statements indicate that 11 studies were included in this systematic study. However, Figure 1 as well as the other sections of the manuscript indicate 8 studies to be included in the study. Check and update accordingly.
  2. There are spelling errors in the manuscript.  The authors should use a spell-checker for the entire manuscript. 
  3. Define all acronyms at first instance of usage. 

We sincerely appreciate the reviewer's thoughtful comments and suggestions, which have helped us improve the clarity and accuracy of our manuscript. Below, we address each point and outline the corresponding revisions made.

Correction of Study Count Discrepancy (Lines 181-184 & Figure 1): We acknowledge the inconsistency regarding the number of included studies. We have carefully reviewed Figure 1 and the related text, ensuring that all sections of the manuscript consistently report the correct number of included studies (8 studies).

Spelling and Typographical Errors: we have thoroughly proofread the manuscript using a spell-checker and manual review to correct all spelling and typographical errors.

Defining Acronyms at First Use: we appreciate the reviewer’s suggestion and have ensured that all acronyms are defined upon their first mention in the manuscript.

Reviewer 3 Report

Comments and Suggestions for Authors

Comment 1: Minor

"This systematic review was performed according to the Preferred Reporting Items 81
for Systematic Reviewand Meta-Analyses (PRISMA) statement (16), b"

You are missing a space

Comment 2: Major

Table 1 - please specifiy what MESH terms where used - and why the keywords were chosen.

Comment 3: Minor

"Citations obtained through the literature search were recorded, duplicates were elimi- 102
nated using EndNote, and titles and abstracts were independently screened by three reviewers 103
(Massimo Giordano, Marina Garofano, AB). Available full-texts, compliant with inclusion and 104
exclusion criteria, detailed below, were also independently reviewed for potentially eligible"

Please consider referring to Figure 1 (the graphic representation under results)  as I was looking for it. 

Commment 3: Major

Why was 2015 January selected as start date? I would think either you all the way back, or 5 years, or 10 years. 9 years seemes an odd choicee?.

"Source: studies published before 2015 and after 28 October 2024"

Comment 4: Major

Table is jumbled up after submission - please correct.

Table 2. Cochrane risk of bias tool for the risk of bias in individual studies

Comment 5: Minor

Please elaborate whether the two primary revieweres agreed on all decisions to include/exclude?

Comment 6: Minor
L245: Vas -> VAs ...

Comment 7: Minor

292: achieved an impressive 292 SUS score of 92/100 with a 75% of us 

Do not use "impressive" and similar please.

Comment 7: Major
I would expect authors to better explain the tools. "Amazon Alexa" paired with the "Buddy Link" software (19) scored ... please elaborate on these, what do they do, what are their approach, and features.
I would expect a table with features compariong the different types of apps, perhaps including which diseaes they cover (at least whether it is a single or many).
Also, please report on the FDA and/or MDR CE approval of these apps. This is a major concern to most clinicians today.

Comment 8:  Major

I would propose that you add more background on VA and CA for context.

Author Response

Comment 1: Minor

"This systematic review was performed according to the Preferred Reporting Items 81
for Systematic Reviewand Meta-Analyses (PRISMA) statement (16), b"

You are missing a space

Correction of Formatting Issues (Line 81): The missing space in "Preferred Reporting Items 81 for Systematic Reviewand Meta-Analyses (PRISMA)" has been corrected.

Comment 2: Major

Table 1 - please specifiy what MESH terms where used - and why the keywords were chosen.

Thank you for your valuable feedback. We did not use MeSH terms in our search strategy because our aim was to ensure a broad retrieval of relevant studies, including those that might not yet be indexed with standardized MeSH terms. Instead, we used free-text keywords to capture variations in terminology and maximize the sensitivity of our search across different databases.

Comment 3: Minor

"Citations obtained through the literature search were recorded, duplicates were elimi- 102
nated using EndNote, and titles and abstracts were independently screened by three reviewers 103
(Massimo Giordano, Marina Garofano, AB). Available full-texts, compliant with inclusion and 104
exclusion criteria, detailed below, were also independently reviewed for potentially eligible"

Please consider referring to Figure 1 (the graphic representation under results)  as I was looking for it. 

Thank you for your suggestion. We have now added a reference to Figure 1 in the relevant section to improve clarity and guide the reader to the visual representation of the study selection process.
Comment 3: Major

Why was 2015 January selected as start date? I would think either you all the way back, or 5 years, or 10 years. 9 years seemes an odd choicee?.

"Source: studies published before 2015 and after 28 October 2024"

Thank you for your observation. The decision to start the search from January 2015 was based on the emergence and increasing adoption of voice assistants in healthcare applications around that time. Prior to 2015, the use of voice assistants in managing noncommunicable diseases (NCDs) was limited, and relevant studies were scarce. This timeframe ensures the inclusion of the most relevant and up-to-date research while maintaining a sufficiently broad window to capture early developments and trends in this rapidly evolving field.

Comment 4: Major

Table is jumbled up after submission - please correct.

Table 2. Cochrane risk of bias tool for the risk of bias in individual studies

Thank you for your feedback. We have carefully reviewed Table 2 and corrected any formatting issues to ensure clarity and proper alignment of the data.
Comment 5: Minor

Please elaborate whether the two primary revieweres agreed on all decisions to include/exclude?

Thank you for your comment, we have changed as follow. Two authors (Massimo Giordano, AB) independently reviewed the titles and abstracts extracted from the database searches to assess their alignment with the inclusion criteria. In cases where they agreed, studies were either included or excluded based on mutual assessment. When discrepancies arose regarding the inclusion or exclusion of a manuscript based on abstract evaluation, these were resolved through discussion and consensus. If consensus could not be reached, a third reviewer (MPDP) was consulted to make the final decision.

Comment 6: Minor
L245: Vas -> VAs ...
Thank you for your comment. We have modified in to the text.

Comment 7: Minor

292: achieved an impressive 292 SUS score of 92/100 with a 75% of us. Do not use "impressive" and similar please.

Thank you for your comment. We have modified in to the text removing the term “impressive”

Comment 7: Major

I would expect authors to better explain the tools. "Amazon Alexa" paired with the "Buddy Link" software (19) scored ... please elaborate on these, what do they do, what are their approach, and features.
I would expect a table with features comparing the different types of apps, perhaps including which disease they cover (at least whether it is a single or many). Also, please report on the FDA and/or MDR CE approval of these apps. This is a major concern to most clinicians today.

Thank you for the comment; we have added the following paragraph in the discussion section. Furthermore, an important aspect of VA implementation in healthcare is their specific functionalities and regulatory approval status. The applications included in this review demonstrate a range of approaches to NCD management, from symptom tracking and behavior change (19, 20, 22) coaching to medication reminders (22, 26) and clinician alerts (22, 24). For example, Amazon Alexa and Echo (19, 25, 31) integrates with wearable devices and mobile health apps to provide personalized lifestyle tracking and health coaching, making it a flexible tool for managing diabetes and obesity and also cardiovascular disease. Similarly, Medly Voice Assistant (24) is specifically tailored for heart failure management, enabling remote symptom tracking, daily patient feedback, and clinician alerts when deterioration is detected. Other applications, such as Vitória and Laura (22, 23), focus on diabetes self-management, offering medication reminders, dietary coaching, and emotional support through conversational artificial intelligence. In addition, the Signal-based virtual assistant studied by Roca et al. (26) is designed to enhance medication adherence in patients with type 2 diabetes and depressive disorder by providing structured reminders and enabling clinician monitoring through a secure messaging platform (Table 10).

Table 10. Comparative Overview of Voice Assistant Applications

Application

Disease

Key Features

AI Capabilities

User Interaction

FDA/MDR CE Approval

Amazon Alexa, Amazon Echo

Diabetes, Obesity, CVD

Personalized coaching, medication reminders, lifestyle tracking

Natural Language Processing (NLP), integration with wearables

Voice-based

Not specified

Medly

CVD

Symptom tracking, clinician alerts, remote monitoring

AI-driven alerts, symptom analysis

Voice + app

Not specified

Vitória

Type 2 Diabetes

Medication adherence, dietary support, behavior change

Behavior Change Techniques (BCTs), patient feedback

Voice-based

Not specified

Laura

Type 2 Diabetes

Emotional support, diabetes education, self-management tools

Avatar-based interactions, NLP

Voice + text

Not specified

Lumen

Mental Health

Cognitive Behavioral Therapy (CBT)-based problem-solving

AI-driven conversation, NLP-based coaching

Voice-based

Not specified

Signal Platform

Type 2 Diabetes, Depressive Disorder

Medication reminders, clinician monitoring, patient self-reporting

AI-assisted chatbot or call based structured messaging

Text or call based

Not specified

Abbreviations: AI, Artificial Intelligence; BCTs, Behavior Change Techniques; CA, Conversational Agent; CBT, Cognitive Behavioral Therapy; CE, Conformité Européenne; CVD, Cardiovascular Disease; FDA, Food and Drug Administration; MDR, Medical Device Regulation; NCD, Non-Communicable Disease; NLP, Natural Language Processing; SaMD, Software as a Medical Device; VA, Virtual Assistant.

Despite the potential of these technologies, regulatory approval remains a significant barrier to clinical adoption. To the best of our knowledge, none of the reviewed voice assistant applications have obtained MDR CE certification or FDA clearance, highlighting the early stage of research in this field. This is further supported by the limited number of studies available, with only eight included in this review, all involving a small patient population. Given the growing emphasis on regulatory compliance for Software as a Medical Device (SaMD), future research should investigate how voice assistants can meet FDA and MDR requirements and explore strategies to ensure their clinical safety and effectiveness.

Comment 8:  Major

I would propose that you add more background on VA and CA for context.

Thank you for the comment; we have added the following paragraph into introduction section. VAs are part of a broader category of conversational agents (CAs), which include Artificial Intelligence driven chatbots capable of engaging in dynamic and interactive conversations. Unlike VAs, which primarily rely on voice commands and responses, CAs can incorporate text-based interactions and more advanced dialogue management to provide tailored healthcare support (5, 12) . These technologies have gained significant attention in healthcare, supporting telemedicine, self-management of chronic diseases, and mental health interventions (13, 14). Integrating VAs and CAs into healthcare systems has the potential to enhance patient engagement, improve access to health information, and support behavioral change strategies, especially for individuals with NCDs (15).

Round 2

Reviewer 3 Report

Comments and Suggestions for Authors

Thank you for the modifications.

Comment 3: Minor

"Citations obtained through the literature search were recorded, duplicates were elimi- 102
nated using EndNote, and titles and abstracts were independently screened by three reviewers 103
(Massimo Giordano, Marina Garofano, AB). Available full-texts, compliant with inclusion and 104
exclusion criteria, detailed below, were also independently reviewed for potentially eligible"

Please consider referring to Figure 1 (the graphic representation under results) as I was looking for it.

Thank you for your suggestion. We have now added a reference to Figure 1 in the relevant section to improve clarity and guide the reader to the visual representation of the study selection process.

Interms of the previous comment 3: Major

Why was 2015 January selected as start date? I would think either you all the way back, or 5 years, or 10 years. 9 years seemes an odd choicee?.

"Source: studies published before 2015 and after 28 October 2024"

Thank you for your observation. The decision to start the search from January 2015 was based on the emergence and increasing adoption of voice assistants in healthcare applications around that time. Prior to 2015, the use of voice assistants in managing noncommunicable diseases (NCDs) was limited, and relevant studies were scarce. This timeframe ensures the inclusion of the most relevant and up-to-date research while maintaining a sufficiently broad window to capture early developments and trends in this rapidly evolving field.

Comment 2.1 (Round 2):

You write there were no papers reported before January 2015 ... but if you did not expand the search - how do you know? I strongly suggest that you double check this - e.g. by adding 2014 - so that you have a 10 year search. Then, you can write in your methods, that no papers were found prior to 2015.

Major comment 2.2:

Also, in terms of the FDA/CE approvals you have now put in the table as I requeted. You write "Not mentioned" for all reported studies ... I find this troubling. While I believe your reporting to be precise, I would strongly recommend that you write a section in the discussion on "Regulatory issues detected". Are they all breaking the law? I would think that virtual assistants are an extreme example of a medical device application. Also, please consider writing the corresponding authors and ask them to explain why they did not report this.   

Author Response

Comment 2.1 (Round 2):

You write there were no papers reported before January 2015 ... but if you did not expand the search - how do you know? I strongly suggest that you double check this - e.g. by adding 2014 - so that you have a 10 year search. Then, you can write in your methods, that no papers were found prior to 2015.

Thank you for your comment. To respond as accurately as possible, we have extended our search to 2014, obtaining the following results:

  • PubMed (2014): 5 studies, none addressing the use of vocal assistants.
  • Scopus (2014): 1 study, a case study.
  • Web of Science (2014): 6 articles, 2 of which were duplicates of PubMed. After screening titles and abstracts, 4 studies remained, but none were clinical trials.

Accordingly, we have updated the PRISMA diagram and revised the references to the search within the text.

Major comment 2.2:

Also, in terms of the FDA/CE approvals you have now put in the table as I requeted. You write "Not mentioned" for all reported studies ... I find this troubling. While I believe your reporting to be precise, I would strongly recommend that you write a section in the discussion on "Regulatory issues detected". Are they all breaking the law? I would think that virtual assistants are an extreme example of a medical device application. Also, please consider writing the corresponding authors and ask them to explain why they did not report this.   

Thank you for your insightful comment. To address this issue thoroughly, we have decided to include the following section in the Discussion: “A key aspect emerging from this systematic review is the lack of explicit mention of regulatory approvals, such as FDA or MDR CE certification, in the included studies. None of the analyzed voice assistants were reported to have undergone regulatory ap-proval processes, raising concerns about their compliance with established medical device regulations. This omission suggests that many of these technologies may not yet meet the safety, efficacy, and data protection standards required for clinical use.

The absence of regulatory approval may be attributed to several factors. First, some of the voice assistants examined in this review are research prototypes or com-mercially available AI-driven tools that have been repurposed for healthcare applica-tions rather than specifically designed as certified medical devices. Second, the regula-tory classification of voice assistants in healthcare remains an evolving area, and many interventions may not yet fall under the category of Software as a Medical Device (SaMD), thus operating in a regulatory gray zone.”

Additionally, we acknowledge the importance of understanding the rationale behind this gap. Therefore, we will consider reaching out to the corresponding authors of the included studies to inquire why regulatory approvals were not mentioned.
